# Graph Principal Flow Network for Conditional Graph Generation

## ABSTRACT

Conditional graph generation is crucial and challenging since the conditional distribution of graph topology and feature is complicated and the semantic feature is hard to capture by the generative model. In this work, we propose a novel graph conditional generative model, termed Graph Principal Flow Network (GPrinFlowNet), which enables us to progressively generate graphs from low- to high-frequency components. Our GPrinFlowNet effectively captures the subtle yet essential semantic features of graph topology, resulting in high-quality generated graph data given a required condition. Extensive experiments and ablation studies showcase that our model achieves state-of-the-art performance compared to existing conditional graph generation models.

## CCS CONCEPTS

• **Do Not Use This Code → Generate the Correct Terms for Your Paper**; *Generate the Correct Terms for Your Paper*; Generate the Correct Terms for Your Paper; Generate the Correct Terms for Your Paper.

## KEYWORDS

Graph generation, conditional graph generation, GFlowNet

**ACM Reference Format:**
Anonymous Author(s). 2018. Graph Principal Flow Network for Conditional Graph Generation. In *Proceedings of Make sure to enter the correct conference title from your rights confirmation emai (Conference acronym 'XX)*. ACM, New York, NY, USA, 11 pages. https://doi.org/XXXXXXX.XXXXXXX

## 1 INTRODUCTION

The task of conditional graph generation is crucial in various domains such as automatic compound discovery, drug design, and more [19, 34, 36, 37, 39]. It requires one to generate graph data conditioned on a specific graph label, e.g. graph property, or category. In general, a graph data with $n$ node is defined as $\mathbf{G} \triangleq (\mathbf{X}, \mathbf{A}, y)$, where $\mathbf{X} \in \mathcal{X} \subset \mathbb{R}^{n \times d}$ is the node feature matrix, $\mathbf{A} \in \mathcal{A} \subset \mathbb{R}^{n \times n}$ is the graph adjacency matrix, and $y \in \mathcal{Y}$ is the graph label. Suppose the target graph distribution of interest is $\mathcal{G}$, and each graph is sampled from $\mathbf{G} \sim \mathcal{G}$, the goal of conditional generation is to learn a generative model $g(\cdot; \cdot) : (\mathcal{X} \times \mathcal{A}) \times \mathcal{Y} \mapsto (\mathcal{X} \times \mathcal{A})$, such that for each graph label $\forall y \in \mathcal{Y}$, the distribution of $g(\epsilon; y)$ approximates the conditional graph distribution $\mathbf{G}|y$ well, where $\epsilon$ is a noise sampled from a known prior $\pi$.

While there is a considerable amount of work and literature dedicated to unconditional graph generation [44], the field of conditional graph data generation is relatively understudied. The main challenge in conditional graph data generation arises from two factors: Firstly, the conditional graph distribution is highly complicated, as the relationship between graph features and topology varies significantly across different graph labels. Secondly, conditional datasets typically consist of fewer data points, leading to a higher demand for the effectiveness of the learning model due to data scarcity.

Admittedly, unconditional models can be transformed into conditional generators by integrating a graph label embedding module, similar to conditional generative vision models [11]. However, existing unconditional graph generation models have inherent limitations, making them unsuitable for unconditional generation. Likelihood-based models, like [6, 24, 35], estimate the likelihood function of the underlying graph data distribution to generate samples. Yet, these models struggle with complex graph structures and computational burdens, making them less suitable for conditional generation. Another class, diffusion-based models like [15, 22, 28], illustrates the state-of-the-art performance in unconditional generation by denoising graph data through reverse diffusion SDE. However, noise insertion in the diffusion process can corrupt semantic information, hindering their ability to capture crucial graph label modes. Hence, diffusion-based models fall short of conditional generation models.

In this work, we leverage graph spectral theory to enhance the learning of subtle yet crucial semantic features. Instead of fitting the likelihood function or recovering the graph from uniform noise, our approach involves progressive graph generation from low to high-frequency components. The low-frequency components correspond to the smallest principal components of the graph Laplacian matrix. This step-by-step approach enables coarse-to-fine learning of the graph. As shown in Figure 1, despite a minor difference in the connection between the clusters, the upper and bottom graphs have distinct graph labels with different topological properties and connectivity. The low-frequency component (on the right) successfully discriminates the connectivity, while the diffusion process results in a completely blurred adjacency matrix, failing to distinguish between them. Thus, the low-frequency component is an ideal starting point for unconditional graph generation as it has a smoother distribution, is easier to learn, and captures subtle yet crucial semantic features of the graph label.

Building upon recent advancements in generative modeling, specifically the Generative Flow Network (GFlowNet) [3, 4], we propose a novel framework called Graph Principal Flow Net (GPrinFlowNet). This framework facilitates conditional graph generation by employing a step-by-step coarse-to-fine approach. In the language of GFlowNet, this progressive generation can be understood as a Markov chain, where the $k$-th intermediate state represents the graph adjacency reconstructed from the $k$ lowest principal components of the graph Laplacian. Unlike GFlowNet, which focuses

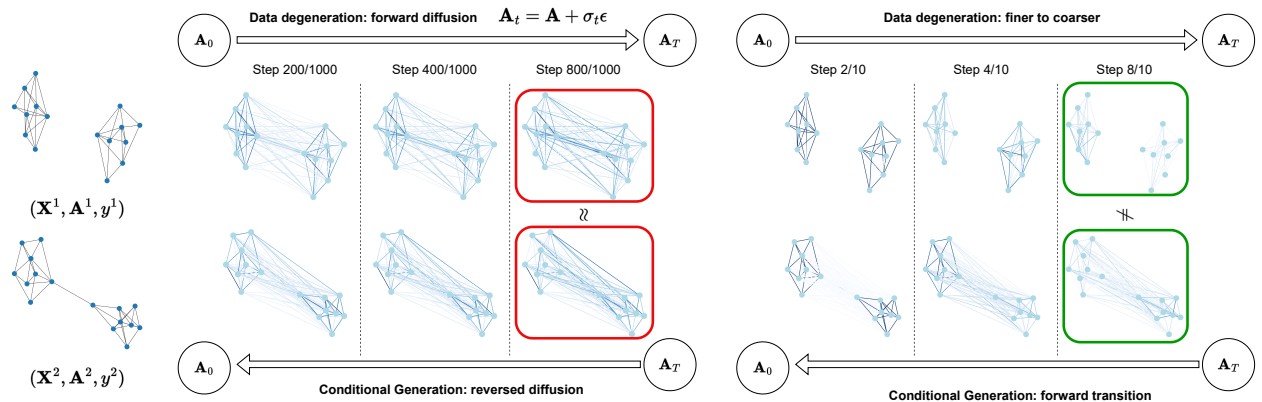

Figure 1: A comparison between the degeneration of graph adjacency across different data degeneration states of diffusion-based models and our GPrinFlowNet. The low-frequency component (green box) proficiently captures the subtle yet crucial patterns that are discriminative to graph labels.

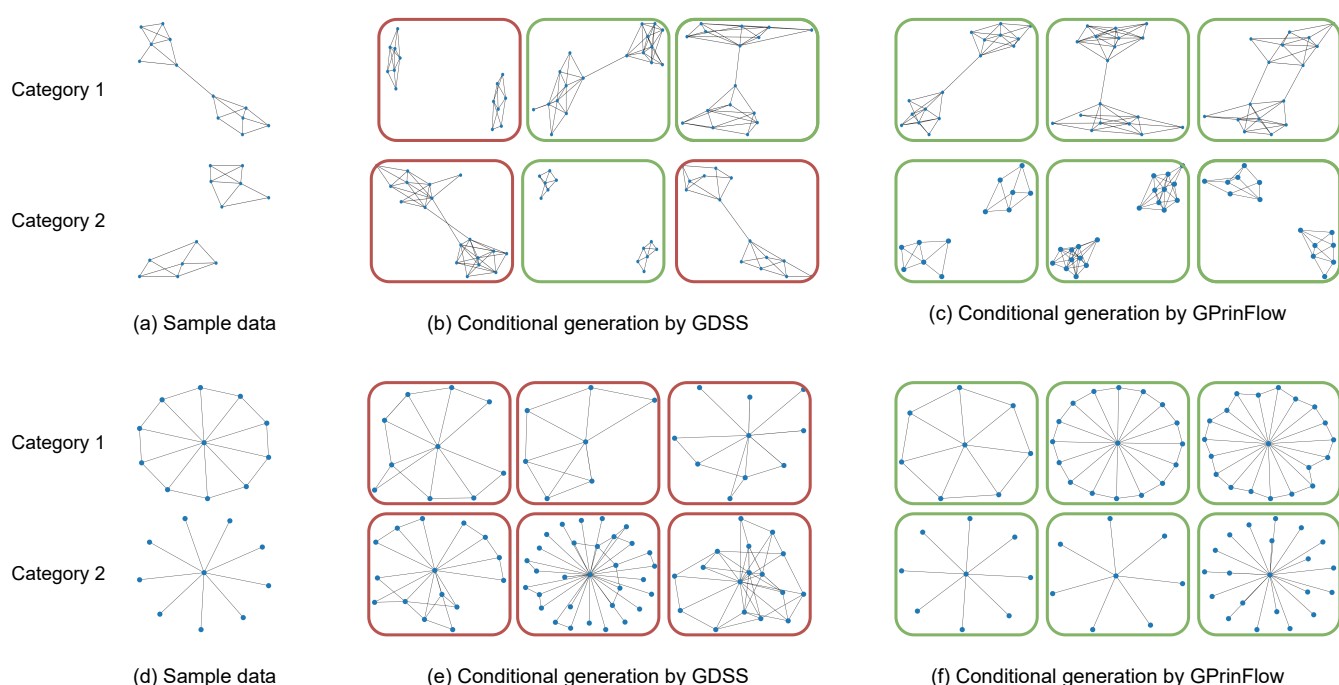

Figure 2: Non-cherry-picked random samples from the testing set as well as samples generated by GPrinFlow (ours) and GDSS [16] under different graph categories. For GDSS, we use the authors' released code to generate the samples. The **red** frame indicates a sample is not correctly generated according to the class category, while the **green** frame indicates a sample is correctly generated.

on discrete probabilistic modeling, GPrinFlowNet simultaneously learns the continuous-valued graph feature and eigenvalues of the graph Laplacian.

In Figure 2, we show a visualization of the conditional generation result of our model and the GDSS, which is a graph generation model based on graph diffusion. On the left side, we show the synthetic graph datasets with 2 categories: community graphs with and without connections between communities, and graphs with and without loops. The visualization demonstrates that our method

can capture the important graph topology information and generate differentiative graphs according to the graph category as a generation condition.

Our contributions are listed as follows.

- We discover the spectral conditional graph generation procedure through an eigenvalue perspective. The procedure of graph generation can be formulated as a process of generating low-frequency terms, which describe the overall properties of the graph, to high-frequency terms, which describe the nuance of the graph properties.

- We propose the Graph Principal Flow Network (GPrinFlow) (Section 4), which has high performance and is computationally efficient for graph conditional generation. Extensive experiments show that our model achieves state-of-the-art performance compared to other types of graph generation methods.

## 2 RELATED WORK

### 2.1 Graph Generation

In recent years, there have been several advanced graph generation strategies proposed, as outlined in [24, 28, 35, 38, 39] and [16]. Among these, GraphRNN [38] and GraphVAE [35] generate nodes and edges sequentially with validity checks, while GAN-based models [6], VAE-based models [24], flow-based models [39], score-based models [16, 28] generate the entire graph in an integrative way and exhibit high computational efficiency due to their node permutation-invariant property, and spectral-based model [22] generates the graph via spectral diffusion by reconstructing the graph eigenvalues through the score-based diffusion.

Different from existing graph generation methods, our proposed GPrinFlowNet networks adopt a novel generation method based on the GFlowNet. Compared to graph diffusion models such as GDSS [16], our model can generate graphs according to the conditional faster and more accurately.

### 2.2 Molecule Generation

Molecule generations are often coherent with graph generation methods, which aim at generating valid meaningful molecules with high efficiency and uniqueness. Molecules inherently adopt a graph-like structure with atoms as nodes interconnected by bonds, represented as edges, making them an optimal input for deep learning models. These molecular graphs are commonly characterized using three matrices: node feature matrix, edge feature matrix, and adjacency matrix. Initially stored in the SMILES format for ease of access, molecules are converted to molecular graphs using tools such as RDKit [17].

Earlier molecule generation approaches utilized sequence-based generative models, representing molecules as SMILES strings. However, these methods often faced challenges from long dependency modeling and had issues with validity since SMILES strings don't guarantee absolute correctness. As a result, recent studies have predominantly adopted graph representations for molecule structures. A variety of graph generative models have been introduced, employing methods like variational auto-encoders [21, 35], generative

adversarial networks [1, 6], normalizing flows [23, 34], and graph diffusion models [12, 16].

### 2.3 Generative Flow Networks

The generative flow networks (GFlowNets) are a stream of generative models which reside at the intersection of reinforcement learning, deep generative models, and energy-based probabilistic modelling. GFlowNets allow neural nets to model distributions over data structures like graphs [3] to sample from them as well as to estimate all kinds of probabilistic quantities (like free energies, conditional probabilities on arbitrary subsets of variables, or partition functions) which otherwise look intractable.

In recent times, there has been a significant uptick in efforts focused on leveraging and enhancing the capabilities of GFlowNet. [8] proposed the DAG-GFlowNet which adopts GFlowNet as an alternative to MCMC for approximating the posterior distribution over the structure of Bayesian networks, given a dataset of observations. [30] proposed the GAFlowNet which applies intermediate rewards by intrinsic motivation to tackle the exploration problem in sparse reward environments. [41] proposed the MLE-GFN based on GFlowNets, which provides a means for unifying training and inference algorithms, and provides a route to shine a unifying light over many generative models. In addition, GFlowNets have been applied to various domains to enhance the model's generation capability on specific domains. For instance, [13] applied GFlowNets on the biological sequence design; [27] adopted GFlowNets for molecule design and drug discovery. In this work, we propose the GPrinFlow, a novel methodology grounded in GFlowNets, revolutionizes with graph conditional generation from a spectral perspective. Our proposed method significantly enhances the conditional generation on various types of graphs.

## 3 PRELIMINARIES

In this paper, we consider graph generation on an undirected and weighted graph $G$, which is an ordered triple $G \triangleq (V, E, W)$, where $V \triangleq \{v_i\}_{i=1}^n$ is a finite set of vertices, $E \subset V \times V$ is the set of edges, and $W : E \mapsto \mathbb{R}_+$ is a weight function that assigns non-negative values to each edge. The adjacency matrix $\mathbf{A} \in \mathbb{R}^{n \times n}$ is then given by $\mathbf{A}[i, j] \triangleq W(v_i, v_j) \cdot \mathbb{I}\{(v_i, v_j) \in E\}$, where $\mathbb{I}\{\cdot\}$ is the indicator function. The degree of node $v$ is $d(v) \triangleq \sum_{u:(u,v) \in E} W(u, v)$ and the degree matrix is $\mathbf{D} \triangleq \text{diag}(d(v_1), ..., d(v_n)) \in \mathbb{R}^{n \times n}$. The normalized Laplacian matrix is $\mathbf{L} \triangleq \mathbf{I} - \mathbf{D}^{-1/2} \mathbf{A} \mathbf{D}^{-1/2}$. $[n]$ denotes $\{1, ..., n\}$.

From the conditional graph generation perspective, we consider the graph generation on $\mathbf{G} \triangleq (\mathbf{X}, \mathbf{A})$, where $\mathbf{X} \in \mathbb{R}^{n \times d}$ represents node features. We focus on the class-conditional graph generation: use class labels $y$ as the generation constraints, and the entire generation objective can be formulated as

$$\hat{\mathbf{G}} = f_\theta(y, \mathbf{G}_0), \qquad (1)$$

where $\hat{\mathbf{G}} \triangleq (\hat{\mathbf{X}}, \hat{\mathbf{A}})$, and $(\hat{\mathbf{X}}, \hat{\mathbf{A}})$ are the generated feature matrix and graph weighted adjacency matrix. $f_\theta(\cdot)$ denotes the learnable generation function, and $\mathbf{G}_0$ denotes a randomly sampled initial state of the graph. The objective is to generate $\hat{\mathbf{G}}|y$ that is similar to $\mathbf{G}|y$ from the test set.

## 3.1 Fundamentals of GFlowNet

Several pivotal developments in generative modeling have led to the emergence of Generative Flow Network (GFlowNet), an advanced model for probabilistic inference [3, 4, 8, 30, 42, 43]. GFlowNet employs a reward function $R(\mathbf{x}) \in \mathbb{R}_+$ to efficiently sample data $\mathbf{x}$ from the data space $\mathcal{X}$. Notably, GFlowNet's sampling protocol follows a Markovian trajectory $\tau = (\mathbf{s}_0, \mathbf{s}_1, ..., \mathbf{s}_n)$, where $\mathbf{s}_0$ is the initial state, $\mathbf{s}_i$ is the $i$-th hidden state, and $\mathbf{s}_n$ is the terminating state with $\mathbf{s}_n = \mathbf{x}$. It's important to note that each hidden state $\mathbf{s}_i$ is derived from the state space $\mathcal{S}$, which may differ from the data space $\mathcal{X}$. Furthermore, the sampling trajectories of GFlowNet, denoted as $\mathcal{T}$, form a Directed Acyclic Graph (DAG), with each node representing a hidden state $\mathbf{s} \in \mathcal{S}$. As highlighted in [4], the sampling process of GFlowNet is governed by the flow function $F(\cdot)$. This function ensures that the measure of incoming trajectories at each hidden state is equal to the measure of outgoing trajectories. Our primary objective is to learn a flow function $F(\cdot)$ such that the total mass of trajectories terminating at $\mathbf{x}$ is proportional to the reward $R(\mathbf{x})$, mathematically expressed as $\sum_{\tau : \mathbf{s}_n = \mathbf{x}} F(\mathbf{x}) = R(\mathbf{x})$.

About the work of Bengio et al. [3], three principal supervisions have been introduced for training GFlowNets. These encompass the flow matching condition [3] as outlined in the same study, the detailed balance condition [4], and the trajectory balance condition [25]. All of these conditions aim to depict the conservation law of flow mass from different levels of granularity. A definition for the flow matching condition is

$$\sum_{\mathbf{s}_{i-1}} F_{\boldsymbol{\theta}}(\mathbf{s}_{i-1}, \mathbf{s}_i) = \sum_{\mathbf{s}_{i+1}} F_{\boldsymbol{\theta}}(\mathbf{s}_i, \mathbf{s}_{i+1}), \qquad (2)$$

where $F_{\boldsymbol{\theta}}(\mathbf{s}, \mathbf{s}') \triangleq \sum_{(\mathbf{s}, \mathbf{s}') \in \tau} F_{\boldsymbol{\theta}}(\tau)$ is the learnable edge flow function. It promotes equality between the masses of incoming and outgoing edge flows. The detailed balanced condition is given by

$$F_{\boldsymbol{\theta}}(\mathbf{s}_i) P_{F,\boldsymbol{\theta}}(\mathbf{s}_{i+1}|\mathbf{s}_i) = F_{\boldsymbol{\theta}}(\mathbf{s}_{i+1}) P_{B,\boldsymbol{\theta}}(\mathbf{s}_i|\mathbf{s}_{i+1}), \qquad (3)$$

where $P_{F,\boldsymbol{\phi}}$ and $P_{B,\boldsymbol{\phi}}$ represent the forward and backward transition probabilities. It aims to achieve equal mass for forward and backward transitions between two consecutive states. To further accelerate the convergence and improve the performance of GFlowNet, the trajectory balance objective extends the detailed balance criterion to an entire trajectory by matching the forward and backward trajectory probabilities $P_{F,\boldsymbol{\theta}}(\tau) \triangleq \prod_{i=0}^{n-1} P_{F,\boldsymbol{\theta}}(\mathbf{s}_{i+1}|\mathbf{s}_i)$ and $P_{B,\boldsymbol{\theta}}(\tau) \triangleq \frac{R(x)}{Z_{\boldsymbol{\theta}}} \prod_{i=0}^{n-1} P_{B,\boldsymbol{\theta}}(\mathbf{s}_i|\mathbf{s}_{i+1})$, where $Z_{\boldsymbol{\theta}}$ is a learnable normalization constant.

## 4 METHODOLOY

### 4.1 Coarse-to-fine Graph Generation Preserves Semantic Information

In this section, we perform empirical analysis to demonstrate that the coarse-to-fine (low- to high-frequency) graph generation curriculum effectively preserves semantic information, i.e. correlation between the graph and its label. While the semantic information attenuates as the graph coarsens, it persists along the whole generation process. On the contrary, the diffusion graph generation curriculum where the semantic information eventually vanishes. Therefore, the coarse-to-fine graph generation turns out to be a highly effective graph conditional generative diagram.

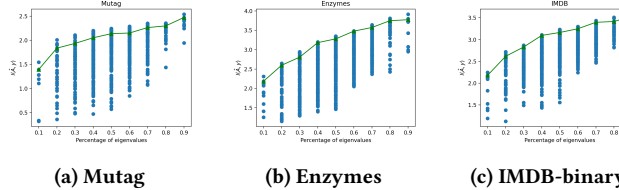

| (a) Mutag | (b) Enzymes | (c) IMDB-binary |
|---|---|---|

**Figure 3: Experiments on the mutual information between different graph frequency components to the graph label. The x-axis is the percentage of eigenvalues used in the dataset, and the y-axis is the mutual information $I(\hat{A}, y)$. Each blue point in the figure represents a result of each frequency component. The green curve shows the mutual information of our proposed method in selecting graph frequency components from low to high.**

As illustrated in Figure 1, the conditional graph generation process can be well achieved by a spectral generation process in a step-by-step eigenvalue generation manner. The underlying research question would be: what kind of step-by-step eigenvalue generation process can enhance conditional graph generation? Intuitively, conditional generation is distinct from unconditional generation, since the condition plays an important role in guiding the generation process. Therefore, a high mutual information between the eigenvalue generation steps to the condition is essential in guiding the model to generate the desired graphs. Different eigenvalue has different effects on the graph connectivity, topology, complexity, etc, and these properties further relate to the graph category. Some eigenvalues have high correlations to the graph categories while some others have low correlations. Thus, in the step-by-step eigenvalue generation process, the eigenvalue generation sequence is important in designing effective conditional graph generation, and our objective is to obtain an effective generation sequence that has high mutual information with the graph labels throughout the generation process.

Starting from this point onward, we conduct a series of experiments to analyze the mutual information of the inner step of our spectral generation process to the generation conditions (i.e. graph labels). We take three commonly used graph classification datasets including Mutag [26], Enzymes [26] and IMDB-binary [26] datasets as examples. We randomly select different percentages of the frequency components (i.e. eigenvalues) ranging among [10%, 20%, 30%, 40%, 50%, 60%, 70%, 80%, 90%], and we follow MINE [2] to compute the mutual information $I(\hat{A}, y)$, where each $\hat{A}$ is the reconstruction using the selected combination of the frequency components. We further elaborate on the experiment details in the Appendix.

We show the corresponding results in Figure 3. Each point in the plot represents an individual experiment that evaluates the mutual information of a randomly sampled graph set at the corresponding percentage of the eigenvalues, and the green lines show selecting the smallest $x\%$ of the eigenvalues, and $x\%$ refers to the value on the x-axis. We can observe that for a fixed $x\%$ eigenvalues selection, selecting the smallest eigenvalues can achieve higher mutual information to the graph labels compared to other selection methods

(i.e. the blue dots in the figure). Such a phenomenon corresponds to the observation shown in Figure 1 and Figure 2, where generating graphs through a coarser-to-finer strategy can enhance the generation performance. Related observations have been found in [40] that small eigenvalues contain important information that is highly graph-topology dependent. Intuitively, generating components that are highly related to the condition instruction (i.e. graph labels) can provide good guidance for the following generation steps. At the coarser level, we generate the low-frequency/low-resolution components that highly correlate to the graph categories, and in the following steps, the generation steps aim to provide fine details at the higher frequencies/resolution, to make the graph more clear and interpretable.

## 4.2 Graph Principal Flow Network

Inspired by the findings in Section 4.1, we propose a low- to high-frequency graph generation algorithm. Assume that each conditional graph instance $\mathbf{G}|y$ is generated by a Markov sample path $\tau \triangleq (\mathbf{s}_0|y, ..., \mathbf{s}_{n-1}|y, \mathbf{G}|y)$, and our ultimate goal is to sample in proportion to the fidelity of $\mathbf{G}|y$. To facilitate the learning process of the transition policies between each successive hidden state $(\mathbf{s}_i|y, \mathbf{s}_{i+1}|y)$, we introduce the following Graph Principal Flow Network. We define the graph Laplacian associated to the adjacency matrix $\mathbf{A}$ as $\mathbf{L} \triangleq \mathbf{D} - \mathbf{A}$, where $\mathbf{D}$ is the diagonal degree matrix defined by $\mathbf{D}[i, i] \triangleq \sum_{j=1}^n \mathbf{A}[i, j]$. We denote the eigen decomposition of $\mathbf{L}$ as $\mathbf{L} = \mathbf{U}\mathbf{\Lambda}\mathbf{U}^\top$, where $\lambda_i \triangleq \mathbf{\Lambda}[i, i]$ is the $i$-th smallest eigenvalue, and $\mathbf{U}[:, i]$ is the corresponding eigenvector.

**Definition 1** (Graph Principal Flow Network). Suppose $d(\cdot, \cdot) : (\mathcal{X} \times \mathcal{A}) \times (\mathcal{X} \times \mathcal{A}) \mapsto \mathbb{R}_+$ is a graph discrepancy score, $\mathcal{T}$ is the Graph Principal trajectory space, where each trajectory $\tau \in \mathcal{T}$ is defined by

$$\tau \triangleq (\mathbf{s}_0|y, ..., \mathbf{s}_{n-1}|y, \mathbf{G}|y), \ \forall y \in \mathcal{Y}, \tag{4}$$

$$\mathbf{G}|y \triangleq (\mathbf{X}, \mathbf{A})|y, \ \mathbf{s}_{n-1}|y \triangleq (\mathbf{X}_{n-1}, \mathbf{A}_{n-1})|y. \tag{5}$$

We assume that the transition between each $(\mathbf{s}_i|y, \mathbf{s}_{i+1}|y)$ follows

$$(\mathbf{s}_{i+1}|\mathbf{s}_i, y) \sim P_{F,\theta}(\mathbf{s}_{i+1}|\mathbf{s}_i; y), \tag{6}$$

$$(\mathbf{s}_i|\mathbf{s}_{i+1}, y) \sim P_{B,\theta}(\mathbf{s}_i|\mathbf{s}_{i+1}; y). \tag{7}$$

Here, $P_{F,\theta}$ and $P_{B,\theta}$ represent the learnable forward and backward transition kernels, which are parameterized by $\theta$. Then a Graph Principal Flow Network is defined as $(d, \mathcal{T}, P_{F,\theta}, P_{B,\theta})$, if there exists a sequence of normalizers $\{Z_i\}_{i=1}^n \subset \mathbb{R}$ such that

$$P_{F,\theta}(\mathbf{s}_{i+1}|\mathbf{s}_i; y)\frac{R(\mathbf{s}_i|y)}{Z_i} = P_{B,\theta}(\mathbf{s}_i|\mathbf{s}_{i+1}; y)\frac{R(\mathbf{s}_{i+1}|y)}{Z_{i+1}},$$

$$R(\mathbf{s}_i|y) \triangleq \exp(-d((\mathbf{X}_i, \mathbf{A}_i), (\mathbf{X}, \mathbf{A}_{(i)}))) \tag{8}$$

Here, $\mathbf{A}_{(i)} \in \mathcal{A}$ is the $i$-th level granularity reconstruction of $\mathbf{A}$, that is

$$\mathbf{A}_{(i)}[k, j] \triangleq (\mathbf{U}\mathbf{\Lambda}_i\mathbf{U}^\top)[k, j] \cdot \delta_{kj}, \ \mathbf{\Lambda}_i \triangleq \mathrm{diag}(\lambda_1, ..., \lambda_i, 0, ..., 0),$$

where the self-loop in $\mathbf{A}$ is omitted. $\mathbf{X}_i$ denotes the feature at each step $i$. At the $i$-th step, our model is expected to generate both the complete feature matrix $\mathbf{X}$ and the $i$-the level granularity reconstruction of $\mathbf{A}$.

GPrinFlowNet stands apart from the standard GFlowNet in two crucial ways: Firstly, each hidden state of GPrinFlowNet resides within a continuous-valued space. Secondly, as demonstrated in equation 8, at the $i$-th intermediate step, the distribution of the generated graph adjacency aligns with the distribution of $\mathbf{A}_{(i)}$ at the corresponding granularity level. This alignment effectively steers GPrinFlowNet to generate conditional graph data incrementally, from lower to higher frequency components.

We propose an effective parameterization and training objective to train a GPrinFlowNet. Specifically, we parameterize the forward and the backward transition kernels as learnable Gaussian distributions, i.e.

$$P_{F,\theta}(\mathbf{s}|\mathbf{s}'; y) \triangleq N(\mathbf{s}; \mu_{F,\theta}(\mathbf{s}', y, \mathbf{U}), \Sigma_{F,\theta}(\mathbf{s}', y, \mathbf{U})), \tag{9}$$

$$P_{B,\theta}(\mathbf{s}|\mathbf{s}'; y) \triangleq N(\mathbf{s}; \mu_{B,\theta}(\mathbf{s}', y, \mathbf{U}), \Sigma_{B,\theta}(\mathbf{s}', y, \mathbf{U})), \tag{10}$$

where the mean and covariance are learned by a multi-layer Graph Convolutional Network (GCN) [7]. Now that the transition probabilities have explicit expression, we can train the GPrinFlowNet by minimizing the following Graph Principal Trajectory Balance objective, which is defined as

$$\mathcal{L}(\theta; \tau) \triangleq \sum_{i=0}^{n-1} \left( \log \frac{Z_{i,\theta} \prod_{j=0}^{i-1} P_{F,\theta}(\mathbf{s}_{j+1}|\mathbf{s}_j; y)}{R(\mathbf{s}_i|y) \prod_{j=0}^{i-1} P_{B,\theta}(\mathbf{s}_j|\mathbf{s}_{j+1}; y)} \right)^2, \tag{11}$$

where the normalizers $\{Z_{i,\theta}\}_{i=1}^n$ are trainable scalars.

In practice, for a labeled training sample $(\mathbf{X}, \mathbf{A}, y)$ with $n$ nodes, we first calculate the graph Laplacian $\mathbf{L}$ and the associated eigenvector $\mathbf{U}$ and eigenvalue matrix $\mathbf{\Lambda}$. Starting from an initial state $s_0 = (\mathbf{X}_0, \mathbf{\Lambda}_0)$, we leverage the forward transition kernel $P_{F,\theta}$ to generate a trajectory of graph samples $(\mathbf{s}_0, ..., \mathbf{s}_n)$, such that $\mathbf{s}_{i+1} \sim P_{F,\theta}(\mathbf{s}_{i+1}|\mathbf{s}_i, y)$. Meanwhile, we compute the backward transition probability using the backward transition kernel $P_{B,\theta}$. Finally, we calculate the graph principal trajectory balance objective by equation 8, and we update the neural parameters $\theta$ via gradient descent. The training process is detailed in Algorithm 1.

## 4.3 Conditional Generation with GPrinFlowNet

With a well-trained GPrinFlowNet, we can efficiently generate high-quality conditional graph data in a maximum of $n$ steps, substantially fewer than the steps required by diffusion-based models. The conditional generation process is detailed in Algorithm 2.

## 5 EXPERIMENTS

## 5.1 Conditional Graph Generation

**Baselines and datasets.** We compare our method with the state-of-the-art graph generation method, including graph diffusion methods such as GDSS [15], EDP-GNN [29]; VAE-based methods such as GraphVAE [35]; auto-regressive models such as GraphAF [34], GraphDF [23], and GraphRNN [38]. Although these existing methods focus on unconditional generation, we effectively modify and extend them for conditional generation by integrating a graph label embedding module, mirroring the approach we employed in GPrinFlowNet. We adopt the AIDS [26] which contains 2 categories, Enzymes [33] which contains 6 categories, and Synthie datasets [9] which contains 4 categories for graph conditional generation.

**Table 1: Generation results on the conditional graph generation datasets. We report the MMD distances between the test datasets and generated graphs. The best results are highlighted in bold (the smaller the better). Hyphen (-) denotes out-of-resources that take more than 10 days or are not applicable due to memory issues.**

| | AIDS | | | | Enzymes | | | | Synthie | | | |
|---|---|---|---|---|---|---|---|---|---|---|---|---|
| | Real, $|V| \leq 95$, $|C| = 2$ | | | | Real, $|V| \leq 125$, $|C| = 6$ | | | | Synthetic, $|V| \leq 100$, $|C| = 4$ | | | |
| | Deg.↓ | Clus.↓ | Orbit↓ | Avg.↓ | Deg.↓ | Clus.↓ | Orbit↓ | Avg.↓ | Deg.↓ | Clus.↓ | Orbit↓ | Avg.↓ |
| GraphRNN [38] | 0.241 | 0.143 | 0.034 | 0.139 | 0.086 | 0.294 | 0.307 | 0.229 | 0.247 | 0.285 | 0.419 | 0.317 |
| GraphAF [34] | 0.197 | 0.093 | 0.026 | 0.105 | 0.058 | 0.174 | 0.156 | 0.129 | 0.137 | 0.176 | 0.302 | 0.205 |
| GraphDF [23] | 0.184 | 0.085 | 0.031 | 0.101 | 0.062 | 0.196 | 0.204 | 0.154 | 1.681 | 1.265 | 0.258 | 1.068 |
| GraphVAE [35] | 0.358 | 0.284 | 0.127 | 0.256 | 1.249 | 0.687 | 0.381 | 0.772 | 1.554 | 1.074 | 0.232 | 0.953 |
| GNF [20] | 0.224 | 0.159 | 0.018 | 0.133 | - | - | - | - | - | - | - | - |
| EDP-GNN [28] | 0.127 | 0.082 | 0.024 | 0.077 | 0.067 | 0.241 | 0.225 | 0.177 | 0.148 | 0.185 | 0.347 | 0.226 |
| GDSS[1] [15] | 0.062 | 0.049 | 0.022 | 0.044 | 0.038 | 0.158 | 0.132 | 0.109 | 0.114 | 0.126 | 0.269 | 0.169 |
| **Ours** | **0.046** | **0.031** | **0.012** | **0.029** | **0.027** | **0.062** | **0.046** | **0.045** | **0.048** | **0.042** | **0.079** | **0.056** |

**Table 2: Conditional generation performance on QM9 with class label $\Delta\epsilon$ - Gap between $\epsilon_{HOMO}$ and $\epsilon_{LUMO}$ (Top), and QM9 with class label $\alpha$ - Isotropic polarizability (bottom). The best results in the first three metrics are highlighted in bold.**

| | Method | VALID w/o check (%) ↑ | NSPDK ↓ | FCD ↓ | VALID (%) ↑ | UNIQUE (%) ↑ | NOVEL (%) |
|---|---|---|---|---|---|---|---|
| Autoreg. | GraphAF | 67.72 | 0.059 | 10.423 | 100.00 | 94.10 | 88.17 |
| | GraphAF+FC | 74.37 | 0.053 | 10.536 | 100.00 | 88.14 | 86.48 |
| | GraphDF | 82.69 | 0.108 | 14.315 | 100.00 | 97.31 | 98.11 |
| | GraphDF+FC | 93.74 | 0.121 | 14.846 | 100.00 | 98.79 | 98.20 |
| One-shot | MoFlow | 91.95 | 0.059 | 8.645 | 100.00 | 98.47 | 94.19 |
| | EDP-GNN | 47.30 | 0.032 | 5.642 | 100.00 | 99.69 | 87.82 |
| | GraphEBM | 8.13 | 0.096 | 10.404 | 100.00 | 97.61 | 96.27 |
| | GDSS | 95.20 | 0.028 | 5.417 | 100.00 | 98.48 | 86.94 |
| | CDGS | 99.41 | 0.021 | 3.326 | 100.00 | 96.79 | 69.73 |
| | GPrinFlowNet (Ours) | **99.72** | **0.012** | **2.798** | 100.00 | 98.87 | 94.71 |

| | Method | VALID w/o check (%) ↑ | NSPDK ↓ | FCD ↓ | VALID (%) ↑ | UNIQUE (%) ↑ | NOVEL (%) |
|---|---|---|---|---|---|---|---|
| Autoreg. | GraphAF | 67.47 | 0.063 | 11.057 | 100.00 | 94.51 | 88.63 |
| | GraphAF+FC | 74.17 | 0.055 | 11.147 | 100.00 | 88.64 | 86.59 |
| | GraphDF | 82.89 | 0.117 | 14.781 | 100.00 | 97.62 | 98.10 |
| | GraphDF+FC | 93.48 | 0.134 | 14.482 | 100.00 | 98.58 | 98.54 |
| One-shot | MoFlow | 91.12 | 0.064 | 8.793 | 100.00 | 98.65 | 94.72 |
| | EDP-GNN | 47.74 | 0.037 | 5.884 | 100.00 | 99.25 | 86.58 |
| | GraphEBM | 8.03 | 0.104 | 10.527 | 100.00 | 97.90 | 97.01 |
| | GDSS | 95.58 | 0.029 | 5.863 | 100.00 | 98.46 | 86.27 |
| | CDGS | 99.44 | 0.023 | 3.741 | 100.00 | 96.83 | 69.62 |
| | GPrinFlowNet (Ours) | **99.74** | **0.013** | **2.925** | 100.00 | 98.85 | 94.72 |

More details of the datasets and the evaluation metric are included in Appendix A.

**Results and analysis.** Following the graph generation evaluation setting [15], for each category, we adopt the same train versus test split ratio as [15]. We measure the maximum mean discrepancy (MMD) to compare the distributions of graph statistics between the same number of generated and test graphs under each category, including the degree, the clustering coefficient, and the number of occurrences of orbits with 4 nodes [15, 38]. Then we report the average of the degree, clustering coefficient, and the number of occurrences among each category in Table 5. We also report the mean MMD as our overall evaluation score under the Avg. column. As shown in Figure 5, our proposed method turns out to be the best performance among the state-of-the-art graph generation baselines.

---

**Algorithm 1** Training GPrinFlowNet

---

**Input:** labeled training data $\mathbf{S}$, the forward and the backward transition networks $P_{F,\theta}$ and $P_{B,\theta}$, the conditional and normalized average feature matrix $\bar{\mathbf{X}}|y$, learning rate $\alpha > 0$.

**Output:** $P_{F,\theta}, P_{B,\theta}, \{Z_{i,\theta}\}_{i=1}^{n}$

**while** not converge **do**
    Sample data $(\mathbf{X}, \mathbf{A}, y) \sim \mathbf{S}$
    $(\mathbf{U}, \mathbf{\Lambda}) \leftarrow \text{EigenDecomp}(\mathbf{A})$
    Initialize $\theta$ and $\mathbf{s}_0 \leftarrow (\mathbf{X}_0, \mathbf{0})$, $\mathbf{X}_0 \sim N(\bar{\mathbf{X}}|y, \mathbf{I})$
    $\mathcal{L}_\theta \leftarrow 0$, $\mathcal{L}_F \leftarrow 0$, $\mathcal{L}_B \leftarrow 0$
    **for** $i = 0$ **to** $n - 1$ **do**
        $\mathbf{s}_{i+1} \sim P_{F,\theta}(\cdot|\mathbf{s}_i; y)$ {Forward transition}
        $\mathcal{L}_F \leftarrow \mathcal{L}_F + \log P_{F,\theta}(\mathbf{s}_{i+1}|\mathbf{s}_i; y)$
        $\mathcal{L}_B \leftarrow \mathcal{L}_B + \log P_{B,\theta}(\mathbf{s}_i|\mathbf{s}_{i-1}; y)$
        $\mathcal{L}_\theta \leftarrow \mathcal{L}_\theta + (\log Z_{i,\theta} - \log R(\mathbf{s}_i|y) + \mathcal{L}_F - \mathcal{L}_B)^2$
    **end for**
    $\theta \leftarrow \theta - \alpha \nabla \mathcal{L}_\theta$
**end while**
**return** $(P_{F,\theta}, P_{B,\theta}, \{Z_{i,\theta}\}_{i=1}^{n})$

---

**Algorithm 2** Conditional Generation with GPrinFlowNet

---

**Input:** training data $\mathbf{S}$, a target label $y$, the forward and the backward transition policy networks $P_{F,\theta}$ and $P_{B,\theta}$, the conditional and normalized average eigenvector matrix $\bar{\mathbf{U}}|y$ and feature matrix $\bar{\mathbf{X}}|y$, a temperature hyperparameter $\sigma > 0$.

**Output:** a plausible conditional graph data $(\widehat{\mathbf{X}}, \widehat{\mathbf{A}})$

Sample $\mathbf{U} \sim N(\bar{\mathbf{U}}|y, \sigma)$, $\mathbf{X}_0 \sim N(\bar{\mathbf{X}}|y, \sigma)$
Initialize $\mathbf{s}_0 \leftarrow (\mathbf{X}_0, \mathbf{0}, \mathbf{U})$
**for** $i = 0$ **to** $n - 1$ **do**
    $\mathbf{s}_{i+1} \sim P_{F,\theta}(\cdot|\mathbf{s}_i, y)$ {Forward transition}
**end for**
$(\widehat{\mathbf{X}}, \widehat{\mathbf{A}}) \leftarrow (\mathbf{X}_n, \mathbf{U}\mathbf{\Lambda}_n\mathbf{U}^\top)$
**return** $(\widehat{\mathbf{X}}, \widehat{\mathbf{A}})$

---

Specifically, compared to GDSS which is one of the state-of-the-art graph generation methods based on the diffusion method, our model achieves a 2.4× and 3.0× lower MMD score, demonstrating the effectiveness of our model.

## 5.2 Conditional Generation on Molecules

Besides generic graph generation, our model can also generate organic molecules. We test our model with a well-known molecule dataset: QM9 [32]. Following previous works [16, 23], the molecules are kekulized by the RDKit library [17] with hydrogen atoms removed. We split the molecules to 2 categories according to $\mu$ - dipole moment, 3 categories according to $\Delta\epsilon$ - gap between $\epsilon_{\text{LUMO}}$ and $\epsilon_{\text{HUMO}}$, and 2 categories according to $\alpha$ - isotropic polarizability. The split is according to the histogram of the corresponding molecule properties shown in Figure 4. We show the details of splitting molecule categories in the Appendix.

Similar to conditional graph generation, we train our model and generate molecules according to the labels under each category. We evaluate the quality of 10,000 generated molecules with **validity and validity w/o check**, **Frechet ChemNet Distance (FCD)** [31],

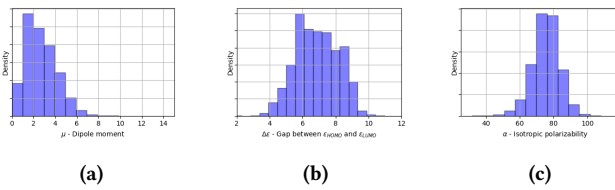

(a)        (b)        (c)

**Figure 4: The histogram of each graph properties: (a). dipole moment (b) gap between $\epsilon_{\text{LUMO}}$ and $\epsilon_{\text{HUMO}}$ and (c) isotropic polarizability of QM9 dataset.**

**Neighborhood subgraph pairwise distance kernel (NSPDK) MMD** [5], Uniqueness [15], and Novelty [15]. FCD computes the distance between the testing and the generated molecules using the activations of the penultimate layer of the ChemNet. (NSPDK) MMD computes the MMD between the generated and the testing set which takes into account both the node and edge features for evaluation. Generally speaking, FCD measures the generation quality in the view of molecules in the chemical space, while NSPDK MMD evaluates the generation quality from the graph structure perspective. Besides, following [16], we also include the **validity w/o correction** as another metric to explicitly evaluate the quality of molecule generation before the correction procedure. It computes the fraction of the number of valid molecules without valency correction or edge resampling over the total number of generated molecules. In contrast, **validity** measures the fraction of the valid molecules after the correction phase.

**Baselines** We compare our model with the state-of-the-art molecule generation models. The baselines include SOTA auto-regressive models: GraphAF [34] is a flow-based model, and GraphDF [23] is a flow-based model using discrete latent variables. Following GDSS [16], we modify the architecture of GraphAF and GraphDF to consider formal charges in the molecule generation, denoted as GraphAF+FC and GraphDF+FC, for fair comparisons. For the one-shot model, we include MoFlow [39], which is a flow-based model; EDP-GNN [28] and GDSS [28] which are both diffusion models; and CDGS [12] which is a diffusion method based on discrete graph structures.

**Results** We show the conditional generation results according to $\Delta\epsilon$ - $\epsilon_{\text{LUMO}}$ and $\epsilon_{\text{HUMO}}$ in Table 2 (top) and $\alpha$ - isotropic polarizability in Table 2 (bottom). GPrinFlowNet achieves the highest performance under most of the metrics. The highest scores in NSPDK and FCD show that GPrinFlowNet can generate molecules that have close data distributions to the real molecules in both the chemical space and graph space. Especially, our model outperforms GDSS and CDGS which are state-of-the-art graph diffusion methods, in most of the metrics, verifying that our proposed GPrinFlowNet is not only suitable for generic graph generation but also advisable for molecule designs.

## 5.3 Unconditional Graph Generation

We also conduct unconditional graph generation experiments with the aforementioned state-of-the-art graph generation methods on synthetic datasets: (1) Community-small [38] ($12 \leq N \leq 20$): contains 100 small community graphs. (2) Enzymes [33]. ($10 \leq$

$N \leq 125$): contains 578 protein graphs which represent the protein tertiary structures of the enzymes from the BRENDA database. (3) Grid [38] ($100 \leq N \leq 400$): contains 100 standard 2D grid graphs. As shown in Table 3, our GPrinFlowNet still achieves state-of-the-art generation results on the unconditional generation task.

## 5.4 Generation Speed Comparison

Furthermore, we compare the graph generation efficiency of some representative aforementioned methods in Table 4. We record and report the graph generation time (in seconds) for generating 100 samples. Our GPrinFlowNet achieves the highest generation speed among all existing methods. The fast speed of our model's generation capability originates from the fast-forward generation process of our model. Specifically, compared to GDSS which is based on the graph Gaussian diffusion, our model achieves 26× and 58× faster, due to the generation mechanism by GPrinFlowNet and the lower generation steps required.

| | Community-small Avg. MMD↓ | Enzymes Avg. MMD↓ | Grid Avg. MMD↓ |
|---|---|---|---|
| DeepGMG [18] | 0.523 | - | - |
| GraphRNN [38] | 0.080 | 0.043 | - |
| GraphAF [34] | 0.133 | 1.073 | - |
| GraphDF [23] | 0.070 | 0.922 | - |
| GNF [20] | 0.170 | - | - |
| GraphVAE [35] | 0.623 | 0.730 | 0.846 |
| EDP-GNN [28] | 0.074 | 0.124 | 0.340 |
| SubspaceDiff [14] | 0.056 | 0.051 | 0.076 |
| WSGM [10] | 0.044 | 0.048 | 0.051 |
| GDSS [15] | 0.046 | 0.046 | 0.062 |
| **Ours** | **0.037** | **0.039** | **0.038** |

**Table 3: Generation results on the unconditional graph datasets. We report the MMD distances between the test datasets and generated graphs. The best results are highlighted in bold (the smaller the better). Hyphen (-) denotes out-of-resources that take more than 10 days or are not applicable due to memory issues.**

| Dataset | GraphAF | GraphDF | EDP-GNN | GDSS | Ours |
|---|---|---|---|---|---|
| Community-small | 357 | $2.47e^3$ | 368 | 72 | **2.7** |
| Enzymes | 596 | $7.58e^3$ | 665 | 128 | **10.2** |
| Grid | $5.83e^3$ | $6.42e^4$ | $7.58e^3$ | $1.75e^3$ | **30.89** |

**Table 4: Graph generation time comparison (in seconds) for generating 100 graphs under the methods' default setting.**

## 5.5 Ablation Studies

We further conduct ablation studies on how the intermediate supervision (e.g. aligning the distribution of the $i$-th intermediate output to the distribution of the reconstruction version of the adjacency matrix at the $i$-th granularity level) affects the performance of GPrinFlowNet. The results presented in Table 5 underscore the

importance of imposing supervision to enable GPrinFlowNet to effectively learn the distribution of the reconstructed graph adjacency matrix at various granularity levels.

| Supervision scheme | AIDS | Enzymes | Synthie |
|---|---|---|---|
| No supervision | 0.037 | 0.075 | 0.086 |
| Supervision per 10 steps | 0.035 | 0.061 | 0.072 |
| Supervision per 5 steps | 0.032 | 0.054 | 0.066 |
| Supervision per 2 steps | 0.030 | 0.049 | 0.058 |
| Supervision in every step | **0.029** | **0.045** | **0.056** |

**Table 5: Ablation studies on the supervision scheme. We report the mean MMD over distributions of degree, clustering coefficient, and the number of orbits, for conditional graph generation.**

To understand the effect of the generation sequence of the graph spectral components on the overall conditional generation performance, we conduct extra experiments to evaluate three different generation procedures: (1) random sequence eigenvalue generation: we apply our GPrinFlowNet to randomly generate the graph eigenvalues, and sort the eigenvalues from smallest to largest in the final step; (2) large-to-small generation: we use our GPrinFlowNet to generate the eigenvalues from largest to smallest, which is a reverse manner of our current proposed method; (3) small-to-large generation: our currently proposed method. We show the corresponding results in Table 6. From the results, we can observe that randomly generating graph eigenvalues is the worst among the three methods. Although generating eigenvalues in a large-to-small manner performs much better than random sequence eigenvalue generation, but it still performs worse than our proposed small-to-large generation procedure.

| Supervision scheme | AIDS | Enzymes | Synthie |
|---|---|---|---|
| Random sequence eigenvalue generation | 0.091 | 0.124 | 0.108 |
| Large-to-small eigenvalue generation | 0.047 | 0.068 | 0.075 |
| Small-to-large eigenvalue generation | **0.029** | **0.045** | **0.056** |

**Table 6: Ablation studies on the eigenvalue generation procedure. We report the mean MMD over distributions of degree, clustering coefficient, and the number of orbits.**

## 6 CONCLUSION

In this paper, we address the challenge of conditional graph generation using the Graph Principal Flow Network (GPrinFlowNet). Through its progressive coarse-to-fine graph generation process, GPrinFlowNet excels at capturing the subtle yet crucial semantic features, making it the state-of-the-art conditional generation model.

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

# A COMPLETE EXPERIMENT RESULTS AND EXPERIMENT DETAILS

We follow the evaluation setting of [15, 38]: we split the data into train/test set according to [15, 38], and sample the same number of graphs as in the test set. Then, we use the maximum mean discrepancy (MMD) to compare the distributions of graph statistics between the same number of generated and test graphs. We follow [15, 38] to measure the distribution difference of degree, clustering coefficient, and the number of occurrences of orbits with 4 nodes. We further average the MMDs and present them in the fourth column under each dataset. We present the complete experiment results for conditional graph generation in Table 7, and unconditional graph generation in Table 8.

The average results of the Enzymes dataset reported in the GDSS original paper is 0.032. However, the best result we can obtain using the author's released code and checkpoint with careful fine-tuning is 0.046.

In our experiments in Figure 3, we randomly select different percentages of the frequency components (i.e. eigenvalues) ranging among [10%, 20%, 30%, 40%, 50%, 60%, 70%, 80%, 90%]. For example, 10% eigenvalues means that we randomly select 10% eigenvalues from the graph and use those eigenvalues to compute $\hat{A}$. Then we follow MINE [2] to compute the mutual information $I(\hat{A}, y)$, and we plot the result $I(\hat{A}, y)$ as one dot under the $x = 0.1$ in the figure.

## A.1 Additional Experiments on Conditional Molecule Generations

For conditional molecule generations, we use $\alpha$ - isotropic polarizability, $\Delta\epsilon$ - the gap between $\epsilon_{HOMO}$ and $\epsilon_{LOMO}$, and $\mu$ - dipole moment as the category indicators. For isotropic polarizability, we set the $\alpha \leq 78$ as one category and $\alpha > 78$ as the second category. For gap between $\epsilon_{HOMO}$ and $\epsilon_{LOMO}$, we set $\Delta\epsilon \leq 6$ as one category, $6 < \Delta\epsilon \leq 8$ as the second category and $\Delta\epsilon > 8$ as the third category. For dipole moment, we set $\mu \leq 3$ one category and $\mu > 3$ as the other category.

We show the molecule conditional generation results with $\mu$ - dipole moment as a graph category indicator in Table 9.

| | AIDS | | | | Enzymes | | | | Synthie | | | |
|---|---|---|---|---|---|---|---|---|---|---|---|---|
| | Real, $|V| \leq 95$, $|C| = 2$ | | | | Real, $|V| \leq 125$, $|C| = 6$ | | | | Synthetic, $|V| \leq 100$, $|C| = 4$ | | | |
| | Deg.↓ | Clus.↓ | Orbit↓ | Avg.↓ | Deg.↓ | Clus.↓ | Orbit↓ | Avg.↓ | Deg.↓ | Clus.↓ | Orbit↓ | Avg.↓ |
| GraphRNN [38] | 0.241 | 0.143 | 0.034 | 0.139 | 0.086 | 0.294 | 0.307 | 0.229 | 0.247 | 0.285 | 0.419 | 0.317 |
| GraphAF [34] | 0.197 | 0.093 | 0.026 | 0.105 | 0.058 | 0.174 | 0.156 | 0.129 | 0.137 | 0.176 | 0.302 | 0.205 |
| GraphDF [23] | 0.184 | 0.085 | 0.031 | 0.101 | 0.062 | 0.196 | 0.204 | 0.154 | 1.681 | 1.265 | 0.258 | 1.068 |
| GraphVAE [35] | 0.358 | 0.284 | 0.127 | 0.256 | 1.249 | 0.687 | 0.381 | 0.772 | 1.554 | 1.074 | 0.232 | 0.953 |
| GNF [20] | 0.224 | 0.159 | 0.018 | 0.133 | - | - | - | - | - | - | - | - |
| EDP-GNN [28] | 0.127 | 0.082 | 0.024 | 0.077 | 0.067 | 0.241 | 0.225 | 0.177 | 0.148 | 0.185 | 0.347 | 0.226 |
| GDSS[1] [15] | 0.062 | 0.049 | 0.022 | 0.044 | 0.038 | 0.158 | 0.132 | 0.109 | 0.114 | 0.126 | 0.269 | 0.169 |
| **Ours** | **0.046** | **0.031** | **0.012** | **0.029** | **0.027** | **0.062** | **0.046** | **0.045** | **0.048** | **0.042** | **0.079** | **0.056** |

Table 7: Generation results on the conditional graph generation. We report the MMD distances between the test datasets and generated graphs. The best results are highlighted in bold (the smaller the better). Hyphen (-) denotes out-of-resources that take more than 10 days or are not applicable due to memory issues.

| | Community-small | | | | Enzymes | | | | Grid | | | |
|---|---|---|---|---|---|---|---|---|---|---|---|---|
| | Synthetic, $12 \leq |V| \leq 20$ | | | | Real, $10 \leq |V| \leq 125$ | | | | Synthetic, $100 \leq |V| \leq 400$ | | | |
| | Deg.↓ | Clus.↓ | Orbit↓ | Avg.↓ | Deg.↓ | Clus.↓ | Orbit↓ | Avg.↓ | Deg.↓ | Clus.↓ | Orbit↓ | Avg.↓ |
| DeepGMG [18] | 0.220 | 0.950 | 0.400 | 0.523 | - | - | - | - | - | - | - | - |
| GraphRNN [38] | 0.080 | 0.120 | 0.040 | 0.080 | 0.017 | **0.043** | 0.021 | 0.043 | | | | |
| GraphAF [34] | 0.18 | 0.200 | 0.020 | 0.133 | 1.669 | 1.283 | 0.266 | 1.073 | - | - | - | - |
| GraphDF [23] | 0.060 | 0.120 | 0.030 | 0.070 | 1.503 | 1.061 | 0.202 | 0.922 | - | - | - | - |
| GraphVAE [35] | 0.350 | 0.980 | 0.540 | 0.623 | 1.369 | 0.629 | 0.191 | 0.730 | 1.619 | **0.0** | 0.919 | 0.846 |
| GNF [20] | 0.200 | 0.200 | 0.110 | 0.170 | - | - | - | - | - | - | - | - |
| EDP-GNN [28] | 0.053 | 0.144 | 0.026 | 0.074 | 0.023 | 0.268 | 0.082 | 0.124 | 0.455 | 0.238 | 0.328 | 0.340 |
| SubspaceDiff [14] | 0.057 | 0.098 | 0.012 | 0.056 | 0.037 | 0.099 | 0.018 | 0.051 | 0.124 | 0.013 | 0.090 | 0.076 |
| WSGM [10] | 0.039 | 0.084 | 0.009 | 0.044 | 0.034 | 0.097 | 0.013 | 0.048 | 0.083 | 0.006 | 0.065 | 0.051 |
| GDSS[1] [15] | 0.045 | 0.086 | 0.007 | 0.046 | 0.026 | 0.102 | 0.009 | 0.046 | 0.111 | 0.005 | 0.070 | 0.062 |
| **Ours** | **0.021** | **0.068** | **0.021** | **0.037** | **0.021** | 0.088 | **0.009** | **0.039** | **0.056** | 0.042 | **0.015** | **0.038** |

Table 8: Generation results on the unconditional graph datasets. We report the MMD distances between the test datasets and generated graphs. The best results are highlighted in bold (the smaller the better). Hyphen (-) denotes out-of-resources that take more than 10 days or are not applicable due to memory issues.

| | Method | VALID w/o check (%) ↑ | NSPDK ↓ | FCD ↓ | VALID (%) ↑ | UNIQUE (%) ↑ | NOVEL (%) |
|---|---|---|---|---|---|---|---|
| Autoreg. | GraphAF | 67.48 | 0.049 | 9.372 | 100.00 | 94.51 | 88.83 |
| | GraphAF+FC | 74.72 | 0.053 | 9.248 | 100.00 | 88.64 | 86.59 |
| | GraphDF | 82.47 | 0.094 | 13.489 | 100.00 | 97.62 | 98.10 |
| | GraphDF+FC | 93.31 | 0.114 | 13.476 | 100.00 | 98.58 | 98.54 |
| One-shot | MoFlow | 91.58 | 0.053 | 8.024 | 100.00 | 98.65 | 94.72 |
| | EDP-GNN | 47.72 | 0.030 | 5.081 | 100.00 | 99.25 | 86.58 |
| | GraphEBM | 8.91 | 0.087 | 9.970 | 100.00 | 97.90 | 97.01 |
| | GDSS | 95.76 | 0.022 | 5.047 | 100.00 | 98.46 | 86.27 |
| | CDGS | 99.17 | 0.017 | 3.024 | 100.00 | 96.83 | 69.62 |
| | GPrinFlowNet (Ours) | **99.79** | **0.011** | **2.627** | 100.00 | 98.64 | 93.75 |

Table 9: Conditional generation performance on QM9 with class label $\mu$ - dipole moment. The best results in the first three metrics are highlighted in bold.

