# OpenReview forum: "Graph Principal Flow Network for Conditional Graph Generation"
_ACM.org/TheWebConf/2024/Conference — TheWebConf24_

### Official Review · Reviewer_gFcf · 2023-11-23

**Novelty:** 6
**Technical Quality:** 6

**Review:**

The authors propose GPrinFlowNet, a novel model that integrates graph spectral theory with Generative Flow Network (GFlowNet) for graph conditional generation. The experimental results on multiple datasets showcase the exceptional performance and computational efficiency of GPrinFlowNet compared to existing approaches. Furthermore, the authors conduct extensive empirical analysis and ablation studies to support the motivation and necessity of their proposed designs.

Overall, the authors introduce an effective method and perform rich experiments to prove its effectiveness on the target task. The writing is clear and coherent, while the obtained results have potential implications for relevant fields.

1. quality: Good
2. clarity: Good
3. originality: Good
4. significance: Good, but might not be related to the Web.

- pros
  a. The writing is clear.
  b. The proposed model achieves state-of-the-art performance compared to other baselines.
  c. Extensive empirical analysis and ablation studies provide evidence for the effectiveness of their designs.
- cons
  a. The relevance of the paper to the Web seems somewhat limited. The authors should provide a more in-depth discussion of the connection between this work and the Web. Currently, no paper from the Web Conf has been cited or discussed.

**Questions:**

I hope the authors can provide a detailed explanation of how their work is related to the Web, which is the biggest concern.

**Ethics Review Description:**

No ethical issues.

**Reviewer Confidence:**

3: The reviewer is confident but not certain that the evaluation is correct

**Scope:**

2: The connection to the Web is incidental, e.g., use of Web data or API

---

### Official Review · Reviewer_f5UQ · 2023-11-26

**Novelty:** 6
**Technical Quality:** 5

**Review:**

This paper is focused on the conditional graph generation problem. It proposes GPrinFlowNet, which is inspired by the Generative Flow Network (GFlowNet). A GFlowNet is a recently proposed generative model that accounts for multiple trajectories for generating an object with an associated reward. Here, the authors model trajectories based on sequences of eigenvectors of the graph Laplacian. The intuition is that low-frequency eigenvectors provide structural information at a coarser level while high-frequency eigenvectors do the same at finer levels. The proposed approach is compared against multiple baselines at both conditional and unconditional generation tasks using three datasets. Results show that GPrinFLowNet outperforms the baselines in terms of several graph statistics while also being more efficient.

Strengths:

+The paper is well written

+The experimental results are promising

+Results for both conditional and unconditional generation are provided

Weaknesses:

-The proposed approach is not compared against GFlowNet

-It is not clear that the baselines considered are the state-of-the-art

-Proposed eigenvector-based generation does not seem to be specific for conditional generation

Detailed comments:

I have enjoyed reading this paper even though there are some parts that are not very clear to me. Here are some details on the weaknesses I have identified:

*GFlowNet vs. GFlowPrinNet: The design of GFlowPrinNet is inspired by GFlowNet but the differences are only described in the following sentences:

“Firstly, each hidden state of GPrinFlowNet resides within a continuous-valued space. Secondly, as demonstrated in equation 8, at the 𝑖-th intermediate step, the distribution of the generated graph adjacency aligns with the distribution of A(𝑖) at the corresponding granularity level.”

These differences should be better clarified both in the description of the method and in the experiments.

*Baselines: The best baseline considered in the paper is GDSS but there are more recent baselines, such as the following:

Vignac et al. DiGress: Discrete Denoising diffusion for graph generation. ICLR’23.
Kong et al. Autoregressive Diffusion Model for Graph Generation. ICML’23

*Eigenvector-based generation: It is unclear how the key contribution of the paper relates to conditional generation. The paper could instead focus on the unconditional generation and show the benefits of the proposed approach against the state-of-the-art.

**Questions:**

1) How is GPrinFlowNet different from GFlowNet?

2) Why better baselines are not considered in the experiments?

3) How are the contributions of the paper related to conditional (vs unconditional) generation?

**Reviewer Confidence:**

3: The reviewer is confident but not certain that the evaluation is correct

**Scope:**

3: The work is somewhat relevant to the Web and to the track, and is of narrow interest to a sub-community

---

### Official Review · Reviewer_WJU9 · 2023-11-26

**Novelty:** 4
**Technical Quality:** 4

**Review:**

What is this paper about：
The paper introduces an approach called Graph Principal Flow Network (GPrinFlowNet) for conditional graph generation. This model enables the progressive generation of graphs, moving from low- to high-frequency components.

what contributions does it make:
1.The paper introduces an approach to spectral conditional graph generation by leveraging an eigenvalue perspective.
2.This paper formulates the graph generation process as a progression from generating low-frequency terms to high-frequency terms.
3.The paper contributes to advancements in the field of conditional graph generation, providing a new model that outperforms existing methods in terms of quality and performance.

the main strengths:
1.The method could capture subtle yet crucial semantic features inherent in graph topology, ensuring the production of high-quality graph data based on specified conditions.
2.Extensive experimental and ablation studies demonstrate that this model surpasses existing conditional graph generation models.

the main weaknesses:
1.The structural organization of the paper requires refinement. Specifically, Section 4.3 is notably brief, and there are concerns about the placement of Algorithm1 and Algorithm2, which appear to be inappropriately situated. Adjustments are recommended to enhance the overall coherence and flow of the paper.
2.The conclusion appears overly concise. It would benefit from further elaboration to provide a more comprehensive summary of the key findings and their implications.
3.The spacing between tables/figures and their surrounding context requires adjustment for better readability and visual clarity.

**Questions:**

1.The CCS CONCEPTS should be concrete.
2.Why not employ likelihood as an evaluation metric to compare the baselines with the proposed method?
3.What is the complexity of the proposed model? The computation of eigenvectors consumes both computing resources and time.

**Reviewer Confidence:**

3: The reviewer is confident but not certain that the evaluation is correct

**Scope:**

4: The work is relevant to the Web and to the track, and is of broad interest to the community

---

### Official Review · Reviewer_ooGK · 2023-11-27

**Novelty:** 5
**Technical Quality:** 2

**Review:**

Quality
Pros: The paper demonstrates rigorous experimentation with detailed results, including a variety of datasets and comparison with several benchmark methods.
Cons: The technical depth of the paper may limit its accessibility to a broader audience. Some sections, particularly the ablation studies, could benefit from more detailed explanations to enhance the reader's understanding.

Clarity
Pros: The paper is structured logically, with clear segmentation of topics and a coherent flow from introduction to conclusion.
Cons: The paper's heavy use of technical jargon and complex mathematical formulations can make it challenging for readers not deeply versed in the field. The figures and tables, while comprehensive, are dense and could be simplified for better clarity.
Originality
Pros: The paper introduces a novel approach to graph generation, advancing the field and offering a new perspective on conditional graph generation tasks.
Cons: While the approach is innovative, the paper does not fully explore the potential limitations or broader implications of the proposed method, leaving room for further exploration.

Significance
Pros: The research addresses a significant challenge in graph generation and offers a solution that could have far-reaching impacts in various applications, including molecular structure prediction.
Cons: The practical applicability of the method in real-world scenarios is not thoroughly explored. The paper could benefit from a discussion on how the approach can be integrated into existing systems or its potential impact on various industries.

Overall Evaluation
The paper presents a significant contribution to the field of graph generation with its novel method. However, it could improve in terms of accessibility and practical applicability. The technical depth, while impressive, may overshadow the broader impact of the research. Future work could focus on simplifying the approach for a wider audience and exploring its practical applications in more depth.

**Questions:**

N/A

**Reviewer Confidence:**

4: The reviewer is certain that the evaluation is correct and very familiar with the relevant literature

**Scope:**

4: The work is relevant to the Web and to the track, and is of broad interest to the community

---

### Official Review · Reviewer_5rVR · 2023-11-27

**Novelty:** 5
**Technical Quality:** 3

**Review:**

This paper considers the conditional graph generation task through a spectral perspective and proposes GPrinFlowNet, a novel methodology grounded in GFlowNet. In particular, GPrinFlowNet adopts the progressive graph generation from low-to-high-frequency components and aligns the distribution of the i-th intermediate output to the distribution of the reconstruction version of the adjacency matrix at the i-th granularity level to enhance the generation performance.

Strength:

1) The spectral conditional graph generation procedure through an eigenvalue perspective is novel and useful.

2) GPrinFlowNet achieves state-of-the-art performance compared to other types of graph generation methods mentioned in this paper.

**Questions:**

1. There are too many typos, even factual errors. Some cases are shown below:

(a) In paragraphs 3 and 4 of the introduction, please distinguish between "conditional generation" and "unconditional generation" in the description.

(b) Where’s Figure 5? I can not find it in the paper.

(c) In Algorithm 1, what is the difference between forward and backward transition kernels? They are both forward transition kernels, according to your description.

(d) GPrinFlow or GPrinFlowNet? The proposed method should be consistent in its abbreviation throughout the paper.

2. In addition to the experimental phenomena, adequate theoretical analysis should be given as to why the coarse-to-fine strategy can work.

3. This paper aims to address the challenge of conditional graph generation. However, all baselines in this paper are originally designed for unconditional generation. Why not compare GPrinFlowNet with existing conditional generation methods?

**Reviewer Confidence:**

2: The reviewer is willing to defend the evaluation, but it is likely that the reviewer did not understand parts of the paper

**Scope:**

3: The work is somewhat relevant to the Web and to the track, and is of narrow interest to a sub-community

---

### Decision · Program_Chairs · 2024-01-22

**Decision:**

Accept

**Comment:**

Aside from one somewhat questionable review (with no responses to rebuttal), the reviews all agree that the paper introduces a novel approach and clearly demonstrated its performance advantages. The are some concerns about the presentation / accessbility of the result, leading to some variations in overall scores.